# The Effect of Colloidal Nano-Silica on the Initial Hydration of High-Volume Fly Ash Cement

**DOI:** 10.3390/ma18122769

**Published:** 2025-06-12

**Authors:** Young-Cheol Choi

**Affiliations:** Department of Civil and Environmental Engineering, Gachon University, Seongnam 13120, Gyeonggi-do, Republic of Korea; zerofe@gachon.ac.kr; Tel.: +82-31-750-5721

**Keywords:** high volume fly ash, colloidal nano-silica, hydration, filling effect, nucleation site, compressive strength

## Abstract

High-volume fly ash cement exhibits drawbacks such as delayed hydration and reduced early-age compressive strength due to the replacement of large amounts of cement with fly ash. In recent years, various studies have been conducted to overcome these limitations by incorporating nanomaterials, such as nano-silica, to promote the hydration of cementitious systems. This study aims to investigate the effect of colloidal nano-silica on the hydration behavior of cement. Cement paste specimens were prepared with varying dosages of colloidal nano-silica to evaluate its influence. To examine the hydration characteristics and mechanical performance, compressive strength tests, isothermal calorimetry, and thermo-gravimetric analyses were conducted. Furthermore, the effect of colloidal nano-silica on the hydration of cement blended with fly ash was also examined. The experimental results revealed that the incorporation of colloidal nano-silica accelerated the hydration reactions in both ordinary and fly ash-blended cement pastes and significantly improved early-age compressive strength. In particular, the 7-day compressive strength of fly ash-blended cement mortar improved by 22.2% compared to the control specimen when 2% colloidal nano-silica was incorporated. The use of colloidal nano-silica appears to be a practical approach for enhancing the early strength of high-volume fly ash concrete, and its broader application and target expansion could contribute to the advancement of a low-carbon construction industry.

## 1. Introduction

In recent years, industries worldwide have shown increasing interest in sustainable development and environmental protection [1]. In particular, the construction sector has focused on achieving carbon neutrality through research and technological innovations. Given that the construction industry accounts for approximately 40% of global CO_2_ emissions, extensive efforts have been made to reduce its environmental impact. A significant portion of the CO_2_ emissions in construction is attributed to the cement industry, prompting growing attention to the use of supplementary cementitious materials (SCMs) for the development of low-carbon cement and concrete [2,3,4,5].

The representative SCMs used in cement and concrete products include fly ash (FA), ground granulated blast furnace slag (GGBS), and silica fume (SF). Among these, FA, a byproduct generated during coal combustion in thermal power plants, is rich in SiO_2_, Al_2_O_3_, and Fe_2_O_3_. It contributes to improved workability due to its spherical particle morphology and exhibits pozzolanic reactivity, which enhances durability and long-term strength development [6]. Notably, high-volume fly ash (HVFA) concrete, in which a large portion of cement is replaced by FA, can reduce CO_2_ emissions by up to 50% [7]. However, its limited early-age strength presents a critical barrier to widespread application, making early strength enhancement a pressing research issue.

To address this limitation, several studies have been conducted. Paya et al. reported that the use of finely ground FA can promote cement hydration and improve early strength through enhanced pozzolanic activity [8]. Riding et al. demonstrated that the incorporation of CaCl_2_ could significantly improve the early-age strength of blended cement systems [9]. Other chemical activators such as sodium sulfate, triethanolamine (TEA), diethanolamine (DEA), and triisopropanolamine (TIPA) have also been studied for their potential to improve early-age strength in HVFA systems [10,11,12,13,14,15].

More recently, there has been growing interest in improving the early-age strength of HVFA concrete using nanomaterials [16,17,18,19,20,21,22,23,24,25,26]. These nanomaterials include nano-silica powder, colloidal nano-silica, titanium dioxide (TiO_2_), and C–S–H seeds. Among the critical considerations when using nanomaterials is dispersion. Improper dispersion can negatively affect strength development [27]. Colloidal nano-silica (CNS) has demonstrated favorable dispersion stability compared to other nanomaterials, offering promising potential for enhancing the properties of cementitious composites. CNS actively participates in cement hydration at the nanoscale, accelerating early hydration, refining the microstructure, and ultimately improving both the strength and durability of concrete [28,29]. Nano-Al_2_O_3_ promotes the hydration reaction and accelerates the initial setting process, thereby contributing to early strength development. However, due to its low pozzolanic reactivity, its effectiveness in enhancing long-term strength is limited. Nano-CaCO_3_ acts as a nucleation site for hydration products and facilitates the early hydration of C_3_S, contributing to the improvement of early strength. Nevertheless, its inherently low chemical reactivity results in minimal enhancement of long-term performance. Typically, the incorporation of nano-CaCO_3_ within the range of 1–3% leads to an increase in early compressive strength, though its effect is generally inferior to that of nano-silica. Nano-TiO_2_ mainly serves photocatalytic and environmental purification functions, with a focus on self-cleaning properties rather than mechanical performance. As such, its influence on early strength is limited, and excessive dosage may even lead to a reduction in strength. In contrast, C–S–H seeds, which are artificially synthesized fine crystals of hydration products, effectively accelerate early hydration and are particularly beneficial for securing early strength in HVFA cement systems where hydration is typically delayed.

CNS is known to contribute to early-age strength improvement through pozzolanic reactivity, the filler effect, and the seeding effect, all of which densify the cement matrix and promote hydration [30]. However, research on the application of CNS in HVFA cement systems remains limited. In particular, studies investigating the interference effect of CNS on the pozzolanic reaction of FA are extremely scarce.

This study investigated the effects of CNS on the hydration behavior of HVFA cement systems. In addition, the influence of CNS on the early-age compressive strength development of HVFA cement mortars was analyzed. To evaluate the hydration characteristics of HVFA cement containing CNS, isothermal calorimetry was conducted to measure the heat of hydration and setting time, while thermogravimetric analysis (TGA) was performed to further assess hydration behavior. Compressive strength tests were carried out at various curing ages to examine both early and long-term strength development.

## 2. Materials and Methods

### 2.1. Materials

Ordinary Portland cement (OPC), FA, and CNS were used as the main raw materials in this study. The chemical oxide compositions of OPC and FA were determined by using an X-ray fluorescence (XRF) spectrometer (Rigaku ZSX Primus II, Rigaku Corporation, Tokyo, Japan) and are presented in Table 1. The densities of OPC and FA were 3.13 g/cm^3^ and 2.34 g/cm^3^, respectively. The particle size distributions of OPC and FA are shown in Figure 1. As shown in Figure 1b, the particle size distribution of FA exhibits a bimodal characteristic, indicating the presence of two distinct peaks in the size distribution curve. The first peak is clearly observed in the range of 10–20 µm, while the second, less pronounced peak appears in the range of 50–60 µm. The average particle diameters of OPC and FA were 19.1 μm and 26.9 μm, respectively.

In this study, the CNS used was a commercial product (YGS-40) manufactured by Company Y (Incheon, Republic of Korea), synthesized through direct oxidation of high-purity silicon. During synthesis, the oxidation of silicon in water generates silicate monomers, which subsequently condense to form silica particles. This process produces spherical, monodisperse particles. The specific gravity and SiO_2_ content of CNS were 1.28 and 40%, respectively. Figure 2 shows an optical image of the CNS solution, which appears as a white, opaque gel. The pH of the CNS solution used in this study ranged from 9.5 to 10.5, indicating a mildly alkaline condition. Under this condition, the CNS particles exhibit a negative surface charge, which increases interparticle repulsive forces and helps maintain stable dispersion. CNS is typically supplied in an aqueous solution form and is based on amorphous silica, which is relatively non-toxic and poses minimal impact on human health and the environment. Silica is generally classified as an inert material, and CNS does not contain volatile organic compounds (VOCs), resulting in negligible emission of hazardous gases during its production and application processes.

SEM images of the CNS particles are shown in Figure 3. The particles exhibit a generally spherical morphology, suggesting isotropic growth during synthesis. Minimal agglomeration is observed, indicating effective stabilization in colloidal form. As seen in Figure 3b, the particle sizes range from 23 nm to 33 nm, with most particles concentrated between 25 nm and 30 nm. The particles are densely packed, but their boundaries remain distinct, indicating that the individual particles are not fused or sintered. The particle surfaces appear smooth, with no evidence of crystalline facets or textures.

The XRD patterns of OPC and FA are presented in Figure 4. As shown in Figure 4a, the main crystalline phases of OPC are C_3_S, C_2_S, C_3_A, and C_4_AF, with clear diffraction peaks also observed for gypsum, calcite, and quartz. In the FA pattern, mullite and quartz appear as the dominant crystalline phases, along with a broad amorphous hump between 2θ ≈ 15–30°, indicative of the pozzolanic potential of the material.

### 2.2. Mixture Proportions

To investigate the effects of CNS on the hydration and mechanical properties of HVFA cement, paste and mortar specimens were prepared according to the mixture proportions shown in Table 2. The primary variables were the replacement levels of FA and CNS. CNS was introduced by replacing a portion of the cement mass with solid-phase nano-silica at 1%, 2%, 3%, and 4% replacement rates. According to previous studies, the incorporation of more than 5% has been reported to adversely affect the strength development of HVFA concrete [31]. FA was incorporated by replacing 0% and 50% of the cement mass to evaluate the interaction effects with CNS. All mixtures were designed with a fixed water-to-binder ratio of 0.5. Standard sand was added at a ratio of 3:1 relative to binder weight. The amount of CNS solution added was adjusted so that the solid content of nano-silica corresponded to the specified replacement ratios, and the water contained in the CNS solution was subtracted from the mixing water to maintain a constant w/b ratio. For compressive strength tests, standard sand conforming to ISO 679 [32] was used. Paste specimens were prepared by following the same mixture proportions as mortars, excluding the sand. Mixing was performed using a forced-type mixer. OPC and FA were added first and mixed for 30 s. CNS solution was then introduced and mixed for 60 s at 100 rpm, followed by an additional 5 min of mixing at 200 rpm.

### 2.3. Test Methods

Initial and final setting times were measured using a Vicat apparatus according to ISO 9597 [33]. The paste mixing was carried out under controlled temperature and humidity, and setting times were recorded using an automatic Vicat device. The early hydration behavior of the cement pastes was evaluated using a TAM Air isothermal calorimeter (TA Instruments, Sollentuna, Sweden). This method provides insight into hydration kinetics and mechanisms during the initial stages. Approximately 4 g of freshly mixed paste (prepared according to Table 2) was sealed in a glass ampoule and placed in the calorimeter chamber. Measurements were conducted under isothermal conditions at 23 ± 0.1 °C for at least 72 h, and the heat flow and cumulative heat release were continuously recorded.

To analyze the hydration products at different ages, TGA was performed using a STA 8122 (SETARAM Instrumentation, Caluire-et-Cuire, France). The paste specimens were hydration-stopped by soaking in isopropanol followed by vacuum treatment for 24 h. Samples were then dried at 40 °C for 24 h, ground into powder, and tested in a nitrogen atmosphere at a heating rate of 10 °C/min up to 1000 °C. Tests were conducted after 3, 7, and 28 days of curing. Compressive strength tests were conducted in accordance with ISO 679 after 3, 7, and 28 days of curing. Immediately after mixing, the mortar was cast into 40 mm × 40 mm × 160 mm prism molds, surface-leveled, and cured at 20 ± 1 °C with >90% relative humidity for 24 h. After demolding, specimens were immersed in a lime-saturated water bath at 20 ± 1 °C until the target age. Each prism was cut at the midpoint, and the compressive strength of each half was measured. An average of six specimens were analyzed for each mix.

## 3. Results and Discussion

### 3.1. Hydration Properties of HVFA Cement Containing CNS

Figure 5 presents the initial and final setting times of the paste specimens, as measured using a Vicat apparatus. As shown in Figure 5a, the incorporation of a high volume of FA significantly increased the initial setting time, with an average increase of approximately 356%. This was attributed to the dilution effect caused by the reduction in C_3_S content due to FA replacement. In both the OPC-based and HVFA-based series, the initial setting time decreased as the CNS content increased. For example, in the OPC-based series, the initial setting time decreased from 5.95 h for OP0 to 3.30 h for OP4, indicating a reduction of 2.65 h with 4% CNS addition. In the HVFA-based series, the initial setting time decreased by 5.46 h when the CNS content increased from 0% to 4%.

Figure 5b shows the final setting times of the specimens, which followed a similar trend to the initial setting times. The final setting time increased by 366% when 50% of the cement was replaced with FA. As with the initial setting time, an increase in CNS content resulted in a consistent reduction in the final setting time in both series. This behavior is attributed to the hydration-accelerating effect of CNS. When incorporated into cementitious materials, CNS interacts with Ca^2+^ ions to promote the formation of C–S–H gel and enhance the rate of hydration [34,35].

Figure 6 shows the heat flow and cumulative heat evolution of OPC-based paste specimens with varying CNS contents. All specimens exhibited a distinct initial hydration peak corresponding to the dissolution and early hydration stages. As the CNS content increased, the time required to reach the second hydration peak decreased, and the magnitude of the peak increased (see Figure 6a). In particular, the second peak appeared at 16.4 h for OP0, whereas for OP4, with 4% CNS, the peak occurred much earlier, at 11.4 h, indicating that CNS effectively accelerated the early hydration of OPC. This can be attributed to the CNS-induced activation of the hydration process around cement particles and the enhancement of nucleation [36].

As shown in the cumulative heat curves in Figure 6b, the other CNS-containing specimens (OP1–OP4) exhibited higher cumulative heat release than OP0 during the initial 36 h. The cumulative heat increased with CNS content, particularly between 6 and 24 h. This suggests that the high specific surface area and reactivity of CNS positively influenced hydration product formation. The results imply that proper incorporation of CNS can contribute to improved early-age strength development and microstructural densification in cement-based materials.

Figure 7 illustrates the heat flow and cumulative heat release of HVFA-based specimens with various CNS dosages. The OP0 specimen, which did not contain FA, was used as a control. As shown in Figure 7a, the specimens with high FA content exhibited delayed hydration behavior compared to OP0, with the main hydration peak appearing later (after approximately 18–25 h). Among these, HVF0 recorded the lowest heat evolution, primarily due to the low initial reactivity of FA and its inert filler effect. However, as the CNS content increased from HVF1 to HVF4, the hydration peak occurred earlier, and the heat flow increased accordingly. These results confirm that CNS positively influences the early hydration of HVFA cement, consistent with the trend observed in Figure 6.

Figure 7b presents the cumulative heat release of HVFA-based specimens. HVF0 exhibited the lowest total heat release. As the CNS content increased, the cumulative heat in the early hydration period also increased. Notably, HVF4 showed the highest cumulative heat over up to 30 h among the FA-containing specimens. Interestingly, HVF2 surpassed HVF4 in terms of total cumulative heat after 36 h and recorded the highest value at 72 h. This behavior is attributed to the nucleation effect of CNS, which enhances the formation and growth of hydration products. Overall, CNS effectively improved the early hydration kinetics and cumulative heat release in HVFA cement systems by mitigating the delayed hydration associated with FA. As a highly reactive amorphous silica, CNS accelerated the formation of hydration products and presented nucleation sites, thereby promoting early hydration and strength development.

### 3.2. Compressive Strength Results

Figure 8 shows the compressive strengths of OPC-based mortars at different curing ages as a function of CNS content. In general, specimens incorporating CNS (OP1–OP4) exhibited higher compressive strength at all ages compared to the control (OP0). At early ages (1 and 3 days), OP2 showed the highest strengths, reaching 13.2 MPa and 30.0 Mpa, respectively. This is attributed to the accelerated hydration and enhanced nucleation of C–S–H promoted by CNS. The positive effect of CNS on strength development persisted through later ages. At 7 and 28 days, OP2 achieved 40.5 Mpa and 49.1 Mpa, respectively, indicating a densified microstructure resulting from ongoing hydration. OP1 and OP3 also consistently outperformed OP0, while OP4, with excessive CNS content, showed slightly reduced strength at certain ages, possibly due to inefficient dispersion or particle agglomeration. Even at 91 days, OP2 maintained the highest strength (56.5 Mpa), likely due to continued pozzolanic reactivity and microstructural densification. OP1 and OP3 also outperformed OP0, whereas OP4 exhibited lower strength compared to the other CNS-modified mixes.

Figure 9 illustrates the compressive strength increase rates of the CNS-modified specimens (OP1–OP4) relative to the control (OP0) across different ages. OP1 exhibited consistent strength gains over OP0, with increases of 2.0%, 7.2%, 5.3%, 3.5%, and 4.6% at 1, 3, 7, 28, and 91 days, respectively. Although the absolute increase was moderate, this suggests that a certain threshold of CNS is necessary to induce stronger reactivity. OP2 achieved the highest strength gains across all ages, with increases of 10.6%, 14.3%, 12.7%, 10.6%, and 11.9% compared to OP0. These results confirm that 2% CNS effectively acts as a nucleation center for C–S–H formation, improving hydration kinetics and microstructural development. OP3 also showed strength gains over OP0, with a slightly lower performance than OP2 at early ages but surpassing OP1 at later ages, suggesting adequate dispersion and sustained reactivity. In contrast, OP4 showed reduced strength at some ages compared to OP0 (−2.8% at 1 day, −0.8% at 7 days, −1.7% at 28 days) and only a marginal gain at 91 days (+2.0%), indicating potential negative effects of excessive CNS, such as agglomeration. Such agglomeration of CNS diminishes its nucleation effect and creates voids between the agglomerates and the matrix, thereby weakening the microstructure and ultimately leading to a loss in strength. According to the study by Elkady et al. [37], the incorporation of 2% nano-silica into cementitious composites resulted in compressive strength improvements of 43% at 1 day and 27% at 28 days compared to the control specimen. However, they reported that exceeding the 2% dosage led to a reduction in strength enhancement due to particle agglomeration.

Figure 10 presents the compressive strength development of HVFA cement mortars with various CNS contents. HVF0, the control without CNS, exhibited compressive strengths of 13.7 Mpa (3 days), 20.1 Mpa (7 days), 36.6 Mpa (28 days), and 48.1 Mpa (91 days), reflecting the typical low early strength of HVFA systems due to the slow reactivity of FA. HVF1 showed improved strength at all ages, achieving 21.6 Mpa at 7 days, 38.9 Mpa at 28 days, and 51.3 Mpa at 91 days. This demonstrates the effectiveness of low CNS dosage in enhancing hydration and densifying the C–S–H matrix. HVF2 exhibited the highest strength among all mixes, with 16.1 Mpa at 3 days, 24.5 Mpa at 7 days, 41.3 Mpa at 28 days, and 55.0 Mpa at 91 days. These results suggest that CNS was optimally dispersed, serving as a nucleation site that promoted both cement hydration and the pozzolanic reaction of FA. HVF3 also outperformed the control and reached 52.5 Mpa in 91 days, though it had a slightly lower strength gain than HVF2, possibly due to reduced dispersion and homogeneity at higher CNS contents. HVF4 showed an improved performance at 91 days (50.6 Mpa) but lower strength at earlier ages, including values similar to HVF0 at 3 and 7 days and a slightly lower strength of 37.3 Mpa at 28 days. The strength increase observed in all specimens at 91 days is attributed to the continued pozzolanic reaction of CNS, which contributes to additional hydration [38].

Figure 11 shows the relative strength gain of CNS-modified HVFA mortars compared to the control (HVF0). HVF1 exhibited modest improvements across all ages except the age of 3 days, where it recorded a 2.6% strength loss. Subsequent increases of 7.4%, 6.1%, and 6.5% occurred at 7, 28, and 91 days, respectively. As observed in the OPC-based mixes (Figure 9), HVF2 with 2% CNS showed the highest strength gains: 16.9% (3 days), 22.2% (7 days), 12.7% (28 days), and 14.3% (91 days). HVF3 followed a similar trend with slightly lower increases than HVF2 but clear improvements over HVF0. HVF4 showed only minor improvement at 3 days (+0.7%) and more noticeable gains at later ages (18.1% at 7 days, 1.7% at 28 days, 5.2% at 91 days). The limited strength gain at 28 days suggests that non-uniform dispersion may have occurred due to excessive CNS. Based on studies on the optimal dosage of nano-silica for improving the early compressive strength of FA-blended cement, Solache et al. reported that a 1% addition of nano-silica resulted in a significant improvement in compressive strength [39]. Furthermore, according to the findings of Supit and Shaikh, in high-volume fly ash (HVFA) cement systems, a 2% nano-silica dosage produced the greatest enhancement in compressive strength at 3 and 7 days, while higher dosages led to a reduction in the strength improvement effect [40]. When applying CNS to actual structures for the purpose of enhancing early strength, it is important to consider that the optimal CNS dosage may vary depending on the mix design conditions of the structure.

### 3.3. Thermogravimetric Analysis Results

Figure 12 presents the TGA curves of OPC-based mortar specimens: the control (OP0), and those with 2 wt% (OP2) and 4 wt% (OP4) CNS additions. In cement paste, mass loss occurs in specific temperature ranges when the paste is exposed to elevated temperatures, primarily due to the evaporation of free water, the dehydration of hydration products, and decarbonation. Therefore, the mass loss in each temperature range can be used to estimate the quantity of hydration products. At the early hydration stage (3 days), all specimens exhibited similar TGA profiles, but OP2 showed a greater mass loss in the 400–500 °C range, corresponding to the decomposition of Ca(OH)_2_. This indicates enhanced hydration and greater portlandite formation in OP2, attributed to CNS-induced hydration acceleration. OP4 showed a similar trend, but also a slightly lower decomposition mass than OP2, suggesting that 2 wt% CNS was more effective during early ages. At 7 days, both OP2 and OP4 exhibited higher decomposition in the Ca(OH)_2_ region compared to OP0, indicating more developed hydration. The difference became more distinct at 28 days, with OP2 displaying the highest mass loss in the Ca(OH)_2_ range, suggesting superior hydration product formation and denser matrix development. Although OP4 still outperformed OP0, it lagged behind OP2, implying that excessive CNS may inhibit optimal hydration due to agglomeration effects. At 91 days, which represents the mature hydration stage, OP2 retained the highest portlandite decomposition, confirming the highest cumulative amount of hydration products. OP4 showed a similar trend, but a relatively lower mass loss, consistent with the compressive strength trends shown in Figure 9.

Figure 13 displays the TGA curves of HVFA mortar specimens with and without CNS over different curing ages. At 3 days, all specimens (HVF0, HVF2, HVF4) showed similar thermal behavior, though HVF2 and HVF4 exhibited slightly greater mass loss than HVF0, indicating enhanced early hydration. At 7 days, HVF2 showed the highest Ca(OH)_2_ decomposition, outperforming HVF0 and HVF4. HVF4, while still better than HVF0, remained less effective than HVF2. At both 28 and 91 days, HVF2 consistently showed the highest mass loss in the portlandite region, demonstrating that CNS acted as a hydration accelerator by promoting the pozzolanic reaction of FA during the active reaction period.

## 4. Conclusions

This study evaluated the effects of colloidal nano-silica (CNS) on the hydration behavior and mechanical properties of OPC and high-volume fly ash (HVFA) cement mortar systems. A comprehensive experimental program, including setting time measurements, isothermal calorimetry, compressive strength testing, and thermogravimetric analysis (TGA), was conducted to assess the influence of CNS at dosages ranging from 1–4 wt%.

The results showed that CNS significantly accelerated the hydration of both OPC and HVFA systems. Setting time was markedly reduced with increasing CNS content, indicating enhanced early hydration due to the nucleation effect. Isothermal calorimetry confirmed that higher CNS contents led to earlier hydration peaks and greater cumulative heat release within the first 36 h. In HVFA systems, which typically exhibit delayed hydration, CNS effectively mitigated this delay and improved early-age heat evolution.

Compressive strength tests revealed that CNS had a positive effect on both early and long-term strength. In OPC-based mortars, 2 wt% CNS (OP2) yielded the highest strength at all ages, attributed to uniform dispersion and effective hydration enhancement. However, excessive CNS content (e.g., 4 wt%) led to reduced performance, likely due to agglomeration and suboptimal pore structure. Similar trends were observed in HVFA mortars, with HVF2 (2 wt% CNS) achieving the highest strength at 91 days among all mixtures. The TGA results supported this, showing higher Ca(OH)_2_ decomposition in 2 wt% CNS specimens, indicating more abundant hydration products. The degree of portlandite formation was strongly correlated with both the calorimetry and mechanical strength results.

In conclusion, CNS is a highly effective admixture for improving the hydration kinetics and mechanical performance of OPC and HVFA mortars. Its use can address the inherent limitations of HVFA systems, particularly their low early-age strength. A CNS dosage of 2 wt% was identified as optimal, achieving a balance between dispersion quality, hydration promotion, and microstructural densification. However, excessive CNS may lead to agglomeration and reduced efficiency, highlighting the importance of precise dosage control, especially in low-carbon HVFA cement systems. CNS is a highly effective material for enhancing the early strength of concrete; however, its practical application in engineering requires precise mix design, the evaluation of material compatibility, appropriate dispersion techniques, and cost–benefit analysis. To ensure the effective use of CNS in large-scale and long-term structures, it is essential to address these practical challenges.

## Figures and Tables

**Figure 1 materials-18-02769-f001:**
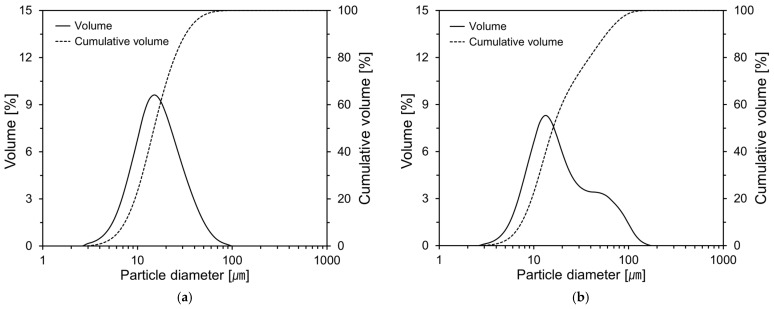
Particle size distributions of raw materials: (**a**) OPC; (**b**) FA.

**Figure 2 materials-18-02769-f002:**
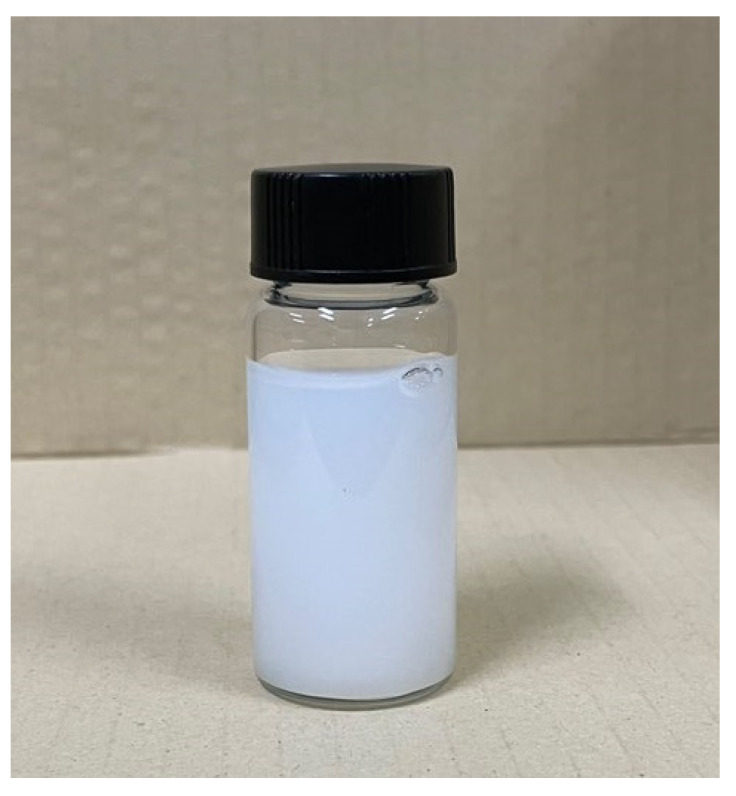
Photograph of CNS (colloidal nano-silica) solution used in this study.

**Figure 3 materials-18-02769-f003:**
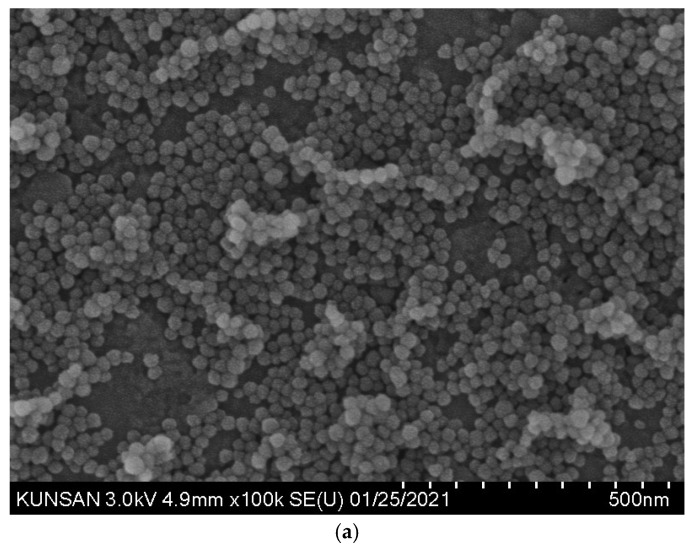
SEM images of CNS (colloidal nano-silica) particles: (**a**) ×100 k; (**b**) ×200 k.

**Figure 4 materials-18-02769-f004:**
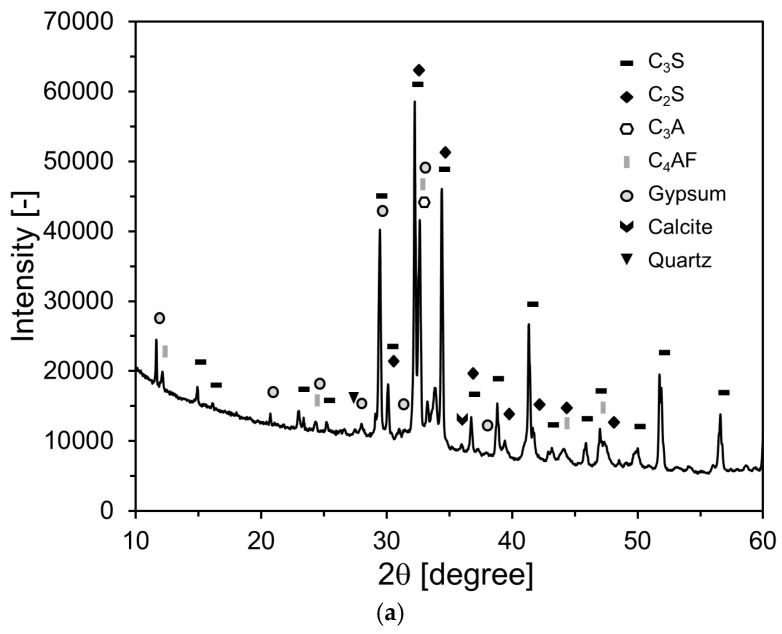
XRD patterns of raw materials: (**a**) OPC; (**b**) FA.

**Figure 5 materials-18-02769-f005:**
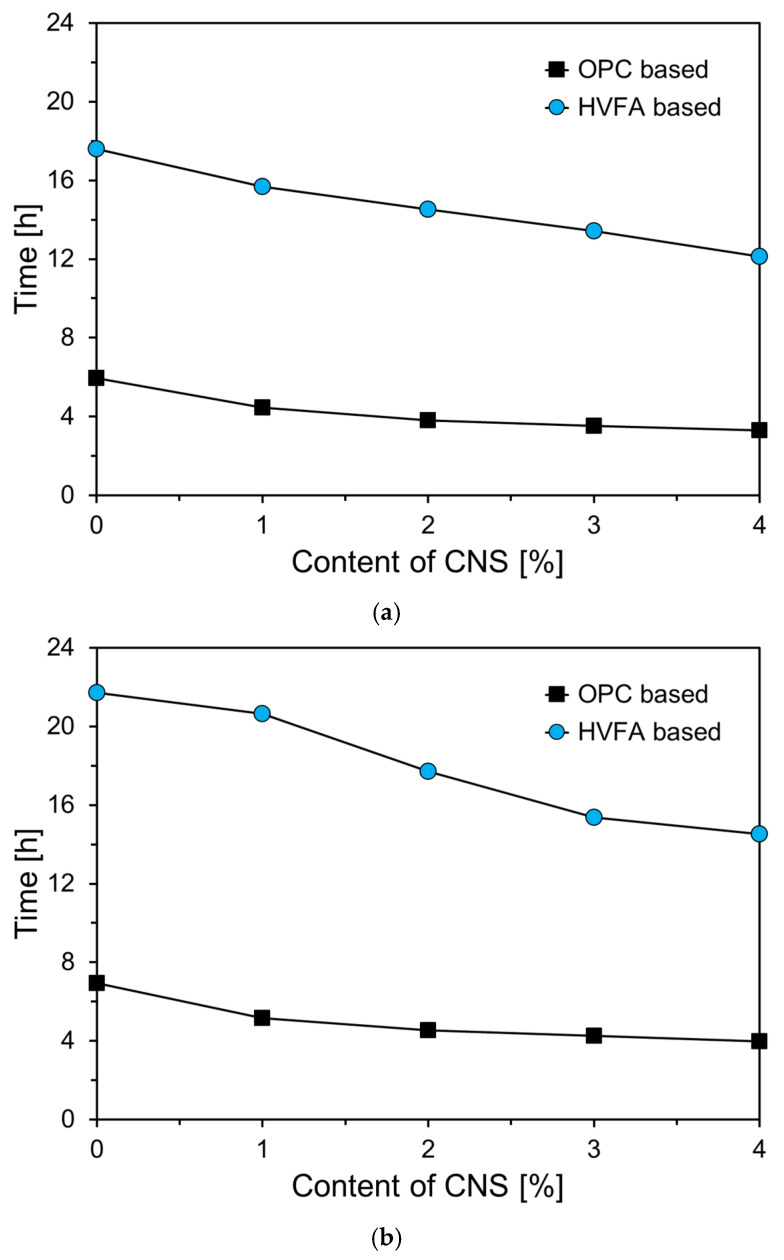
Setting times of specimens: (**a**) initial setting time; (**b**) final setting time.

**Figure 6 materials-18-02769-f006:**
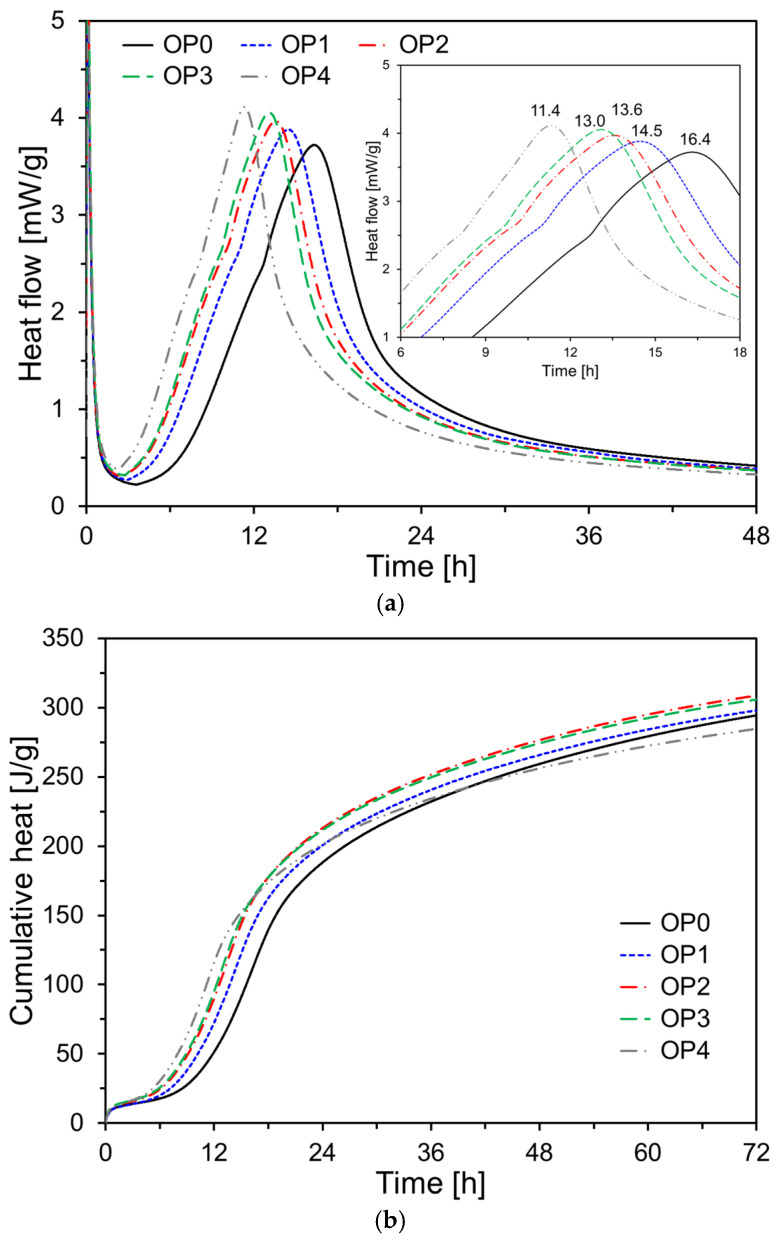
Heat of hydration results of OPC-based specimens: (**a**) heat rate; (**b**) cumulative heat.

**Figure 7 materials-18-02769-f007:**
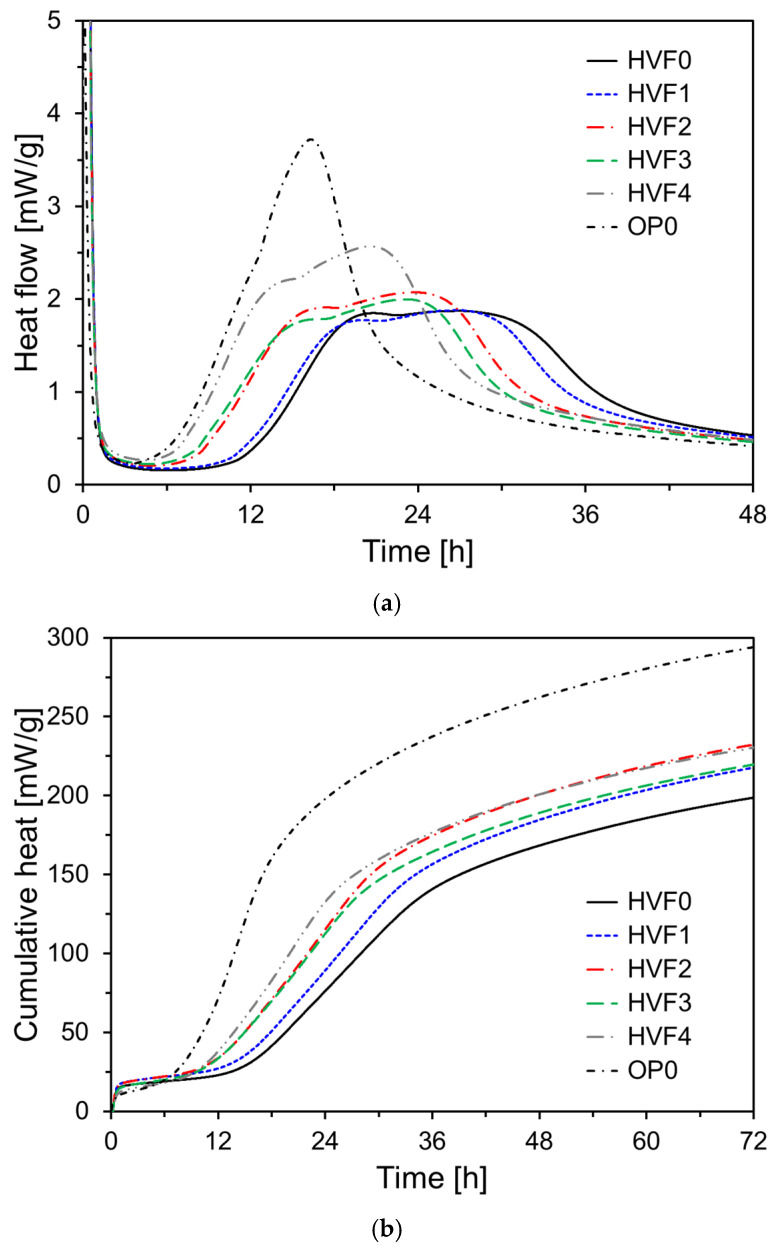
Heat of hydration results of HVFA-based specimens: (**a**) heat rate; (**b**) cumulative heat.

**Figure 8 materials-18-02769-f008:**
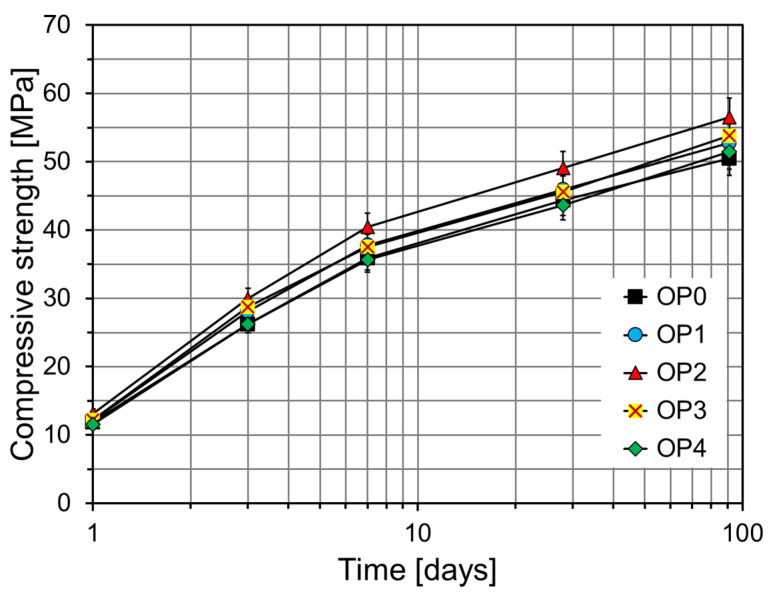
Compressive strength results of OPC-based specimens.

**Figure 9 materials-18-02769-f009:**
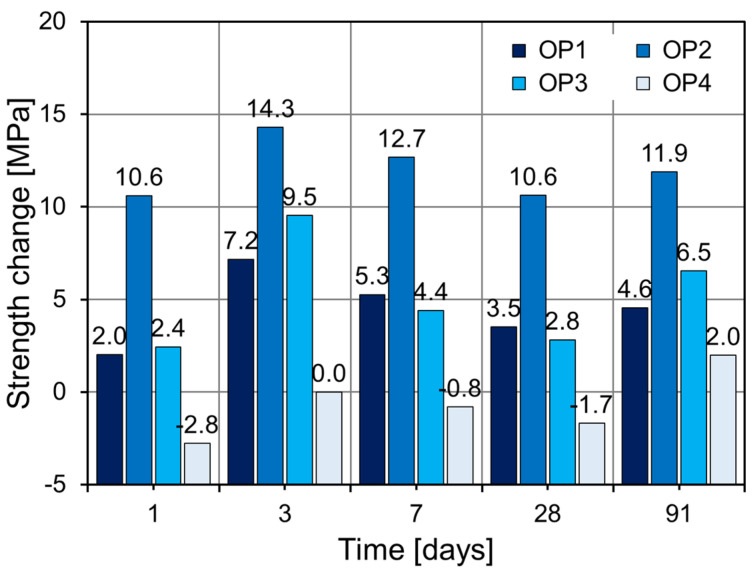
Strength changes in OPC-based specimens.

**Figure 10 materials-18-02769-f010:**
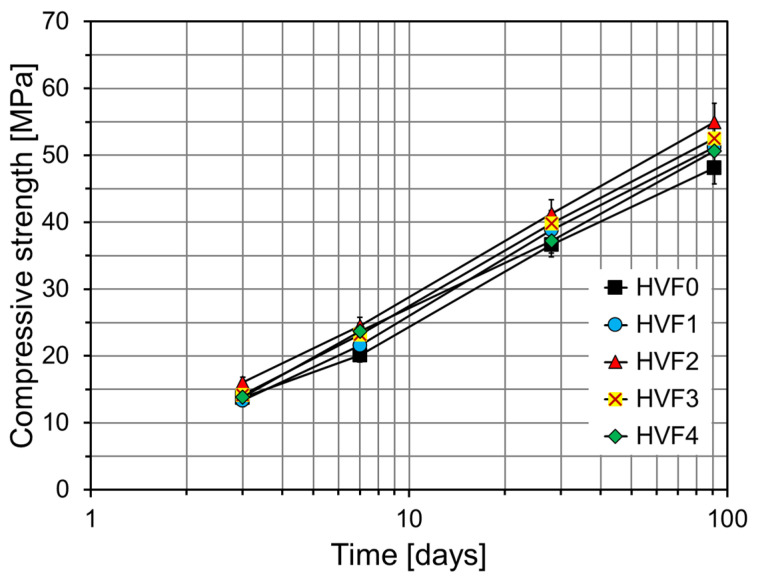
Compressive strength results of FA-based specimens.

**Figure 11 materials-18-02769-f011:**
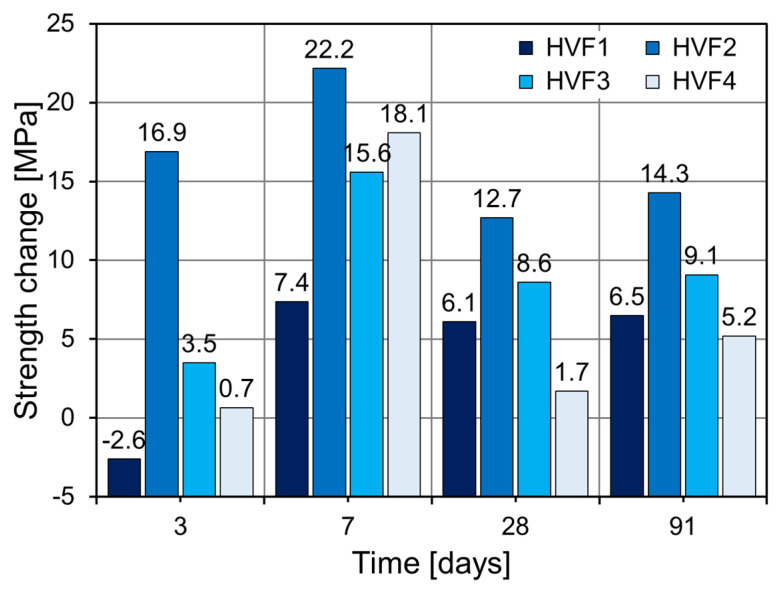
Strength changes in FA-based specimens.

**Figure 12 materials-18-02769-f012:**
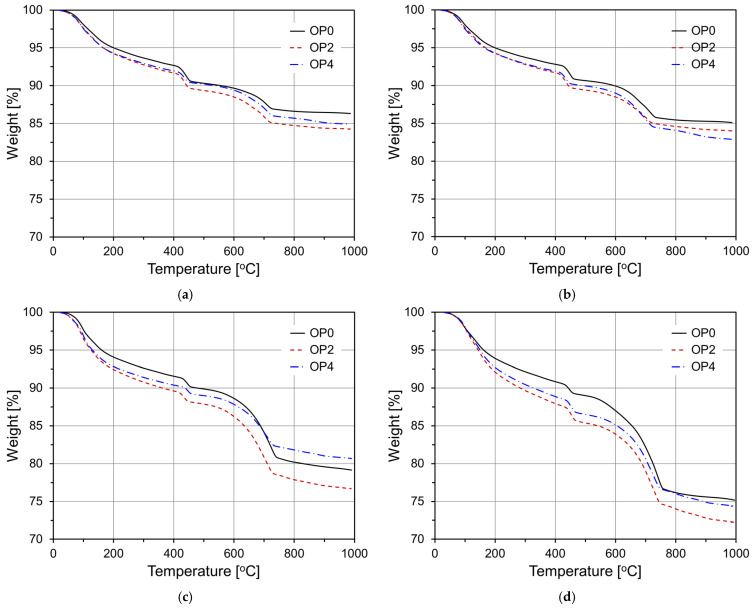
Thermogravimetric analysis curves of OPC-based specimens: (**a**) 3 d; (**b**) 7 d; (**c**) 28 d; (**d**) 91 d.

**Figure 13 materials-18-02769-f013:**
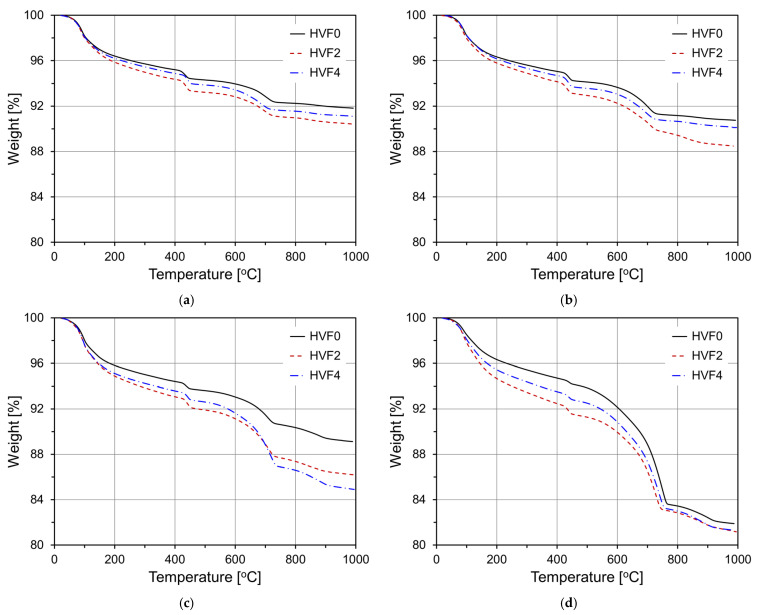
Thermogravimetric analysis curves of FA-based specimens: (**a**) 3 d; (**b**) 7 d; (**c**) 28 d; (**d**) 91 d.

**Table 1 materials-18-02769-t001:** Chemical oxide compositions of OPC and FA.

	Chemical Oxide Compositions (wt%)								
	SiO_2_	Al_2_O_3_	Fe_2_O_3_	CaO	MgO	K_2_O	Na_2_O	SO_3_	TiO_2_
OPC	21.8	4.7	3.3	63.2	3.2	1.7	0.3	2.4	0.3
FA	55.6	22.2	7.8	8.6	2.4	1.2	1.9	2.1	1.0

**Table 2 materials-18-02769-t002:** Mixture proportions in paste and mortar specimens of HVFA cement.

Specimens	W/B (-)	OPC (g)	FA (g)	CNS-Solid (g)	Sand (g)
OP0	0.5	100	-	0.0	300
OP1		99	-	1.0	
OP2		98	-	2.0	
OP3		97	-	3.0	
OP4		96	-	4.0	
HVF0		50	50	0.0	
HVF1		49	50	1.0	
HVF2		48	50	2.0	
HVF3		47	50	3.0	
HVF4		46	50	4.0	

## Data Availability

The original contributions presented in this study are included in the article. Further inquiries can be directed toward the author.

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
