# Peer review of "The Effect of Colloidal Nano-Silica on the Initial Hydration of High-Volume Fly Ash Cement"

_materials, 2025, doi:10.3390/ma18122769_

Round 1
Reviewer 1 Report
Comments and Suggestions for Authors
- The abstract mentions that "the incorporation of CNS significantly improves the early compressive strength", but it could explicitly state the specific numerical increase in strength to enhance the credibility of the results.
- Although the introduction mentions the improvement effect of nano-materials (such as nano-silica) on the early strength of high-volume fly ash (HVFA) cement, the comparative analysis of different nano-materials is not in-depth enough.
- The introduction emphasizes the low-carbon advantage of the HVFA system, but the environmental impact and cost-effectiveness of CNS are not discussed throughout the paper.
- The CNS dosage is set at 1% to 4%, but the reason for choosing this range is not explained, and the influence of higher dosages (such as 5% to 6%) is not considered.
- Key parameters affecting the hydration reaction, such as the pH value and surface charge of CNS, are not mentioned.
- The compressive strength test mentions "an average of six samples", but the standard deviation or error bars are not provided, making it impossible to assess the reliability of the data.
- The conclusion states that "2 wt% CNS is the optimal dosage", but it does not discuss the applicability of this conclusion (such as the influence of different cement types and curing conditions), nor does it mention the potential challenges of CNS in practical engineering applications.
- The paper speculates that the strength decrease at 4% CNS is due to "insufficient dispersion or agglomeration", but could SEM images or particle size analysis data be provided to directly prove the agglomeration phenomenon?
Reviewer 2 Report
Comments and Suggestions for Authors
The paper entitled Effect of colloidal Nano-silica on the initial hydration of high-volume fly ash cement studies the effect of CNS on the behavior of the properties of OPC and HVFA cement mortars. The paper is very clearly presented and well detailed. From my point of view, this work is a strong candidate to be published in the journal Materials. I have detected some minor corrections that the author should consider to improve the work, which I list below:
The last sentence of the introduction should be removed as it makes a conclusion of the work that should not be exposed in this part of the paper (commonly called a spoiler).
Change the name of section 2 to Experimental.
In section 2.1. Materials the author should indicate the brand and model of the XRF equipment used.
The wording of the last sentence is a bit confusing; it does not reflect the bimodality of the FA particle size distribution.
Change Figure 2 shows the optical image by removing optical.
Title of Figure 2 remove "is".
Change 23.0 to 33 nm, either put 23 or use one decimal place like 22 nm. Also, it is more correct to put the units for both magnitudes instead of only the last one.
Section 2.2. Mixture Proportions:
The author states that 1%, 2%, 3%, and 4% were used; the question is clear, why is the maximum 4% and not 5%, for example? The author should give a reason to clarify the choice of this range of CNS percentages used.
Include the reference for ISO 679 standard.
The title of Table 2 I would enlarge by putting ... HVFA cement, paste and mortar specimens.
Section 2.3. Test methods:
Include the references for ISO 9597 and ISO 679 standards.
Section Results and discussion
On page 7, put a space before (see....
Page 10, move the figure title to the previous page.
At the end of the first paragraph, it would be necessary to answer so what?, I miss a brief explanation of the negative effect observed with an excess of CNS.
Page 13. Move the title up to page 12.
Reviewer 3 Report
Comments and Suggestions for Authors
The manuscript addresses a topic of notable relevance and growing interest within the discipline, and the authors are to be commended for their effort in developing a study with clear potential for scholarly contribution. The content is engaging and aligns well with current research directions. However, to fully meet the expectations of an international scientific audience, several enhancements are required. These include refinements in the manuscript’s structural organization, improved clarity in the presentation of central ideas, and a more cohesive narrative that effectively connects technical findings to broader scientific and practical contexts. Addressing these areas will improve both the accessibility and the overall impact of the work. To support the authors in strengthening the manuscript, a set of detailed, constructive comments has been provided within the attached PDF, covering both substantive content and editorial aspects.
1 - I recommend that the authors consider revising the title of the manuscript to make it more engaging and impactful. A more concise and compelling title could help attract greater interest from potential readers and better reflect the novelty or significance of the study’s findings. The title is the first element of the paper that readers encounter, and it plays a crucial role in shaping their initial impression. A well-crafted title can significantly enhance the visibility of the work, increase its citation potential, and make it more accessible to a broader audience, including interdisciplinary researchers. Therefore, investing effort in selecting a more appealing and precise title could substantially benefit the overall dissemination and reception of the research.
2 - I recommend that the authors revise the abstract to place greater emphasis on the applications and broader implications of their research. While the current version outlines the methodology and key findings, it would benefit from a clearer articulation of how the results can be applied in practice or how they contribute to advancing the field. Highlighting the potential impact of the study can make the abstract more compelling and informative, particularly for readers who are scanning for relevance to their own work or interests. By explicitly stating the practical, theoretical, or societal implications, the authors can enhance the perceived value and significance of their contribution.
3 - I suggest that the authors limit the use of acronyms in the abstract, unless they are widely recognized and essential for clarity. The abstract should be accessible to a broad readership, including those who may not be familiar with field-specific terminology. Excessive or unexplained acronyms can hinder comprehension and reduce the overall readability of the abstract. Given that the abstract often serves as the primary means by which readers decide whether to engage with the full article, ensuring clarity and accessibility at this stage is particularly important. Therefore, avoiding or minimizing acronyms—unless absolutely necessary—would improve the overall communication and impact of the work.
4 - I encourage the authors to consider adding a few additional keywords to enhance the visibility and discoverability of the article. Keywords play a crucial role in how papers are indexed by major academic search engines and databases, and they significantly influence the paper’s accessibility to researchers conducting literature searches. By including well-chosen, relevant keywords—particularly those that reflect the core themes, methods, or applications of the study—the authors can improve the chances that their work will reach the appropriate audience and be cited more widely. Expanding the keyword list slightly would therefore be a simple yet effective way to increase the impact of the publication.
5 - While the introduction provides a clear overview of the relevance of sustainable practices in the construction industry and effectively motivates the use of supplementary cementitious materials (SCMs), it could benefit from a more structured articulation of the knowledge gap the present study aims to address. Although the limitations of high-volume fly ash (HVFA) concrete are mentioned, particularly in terms of early-age strength, the transition to nanomaterial solutions—specifically colloidal nano-silica (CNS)—could be better framed in the context of existing challenges and unresolved questions in the literature. Explicitly highlighting what remains insufficiently understood or optimized regarding CNS application would help clarify the novelty and necessity of the current investigation. Strengthening this aspect would provide a sharper research focus and improve the overall scientific positioning of the study.
6 - at the beginning of the introduction, where authors state "[...] In recent years, industries worldwide have shown increasing interest in sustainable development and environmental protection [...]", this sentence lacks of citation to justify the concept expressed. In order to better contextualize these concepts in the recent international literature scenario, authors should cite here the following work, and also others by the authors, dealing exactly with sustainable development and environmental protection also regarding industries:
-Tomassi, A., Falegnami, A., Meleo, L. and Romano, E., 2024. The GreenSCENT Competence Frameworks. In The European Green Deal in Education (pp. 25-44). Routledge.
7 - I suggest that the authors revise the caption of Figure 1 to make it more descriptive and self-explanatory. Currently, the caption provides limited guidance, which may hinder the reader's ability to fully understand the figure without referring extensively to the main text. A well-written caption should clearly explain the content and purpose of the figure, define any symbols or abbreviations used, and, where appropriate, highlight the key message it conveys. Enhancing the clarity and completeness of the caption will improve the figure’s standalone value and make the paper more accessible to readers who may scan figures independently of the full text.
8 - I recommend that the authors consider enlarging Figure 2 to improve its readability and visual impact. As it currently stands, the figure appears too small, making it difficult to clearly distinguish key elements, labels, or data points. Ensuring that figures are easily legible is essential for effective communication of the results, especially for readers who rely on visual summaries to grasp complex information. Increasing the size of Figure 2 would enhance its clarity, facilitate interpretation, and strengthen the overall presentation of the paper.
9 - I also suggest that the authors revise the caption of Figure 2 to make it more detailed and self-explanatory. At present, the caption lacks sufficient information to allow readers to fully understand the figure without referring back to the main text. A clear and comprehensive caption should describe what is being shown, explain any symbols, colors, or abbreviations used, and summarize the key takeaway. Improving the caption in this way would enhance the figure’s standalone interpretability and contribute to a more accessible and reader-friendly presentation of the research.
10 - I recommend that the authors, where possible, enlarge the figures to improve their readability and visual impact, and ensure that all figure captions are made more detailed and self-explanatory. Larger images would facilitate clearer interpretation of data, enhance the visual quality of the paper, and ensure that important features are not overlooked due to scaling limitations. At the same time, captions should provide sufficient context, describe all relevant elements of the figures (including symbols, abbreviations, and color codes), and clearly convey the main message. Improving both the size and descriptive quality of the figures would significantly enhance the overall clarity and accessibility of the manuscript.
11 - While the results and discussion section presents a comprehensive and data-rich analysis of the hydration behavior and strength development in HVFA cement systems containing CNS, the narrative would benefit from more critical interpretation and synthesis of the findings. At times, the discussion leans toward a descriptive summary of observed trends without sufficiently connecting them to underlying mechanisms or positioning them in the context of existing literature. For example, while the acceleration of hydration and strength development with CNS addition is clearly shown, a deeper discussion of the limits of CNS effectiveness—such as the agglomeration effects observed at higher dosages—and how these align or contrast with previous studies could enhance the scientific depth. Furthermore, a clearer delineation of the practical implications of these findings (e.g., optimal CNS dosage thresholds for industry application) would strengthen the relevance of the research. Including a brief discussion on potential long-term durability impacts would also broaden the scope of the analysis.

Round 2
Reviewer 3 Report
Comments and Suggestions for Authors
Dear Authors,
I am pleased to inform you that I have completed my review of the revised manuscript. It is evident that the changes made in response to the reviewers' comments have significantly enhanced the clarity and depth of your paper. The efforts you have put into addressing the concerns and suggestions have not only improved the manuscript but have also augmented its contribution to the field.
In light of the substantial improvements made, I believe that the manuscript is now well-prepared for publication. The revisions have effectively strengthened the arguments, enriched the data presentation, and refined the overall narrative, thereby solidifying its scholarly value.
Thank you for your diligence and commitment to enhancing your work. I look forward to seeing your research published and contributing to ongoing discussions in your area of expertise.
Best regards